# Preparation and Antioxidant Activity of New Carboxymethyl Chitosan Derivatives Bearing Quinoline Groups

**DOI:** 10.3390/md21120606

**Published:** 2023-11-24

**Authors:** Linqing Wang, Rui Guo, Xiaorui Liang, Yuting Ji, Jingjing Zhang, Guowei Gai, Zhanyong Guo

**Affiliations:** 1Key Laboratory of Coastal Biology and Bioresource Utilization, Yantai Institute of Coastal Zone Research, Chinese Academy of Sciences, Yantai 264003, China; linqingwang@yic.ac.cn (L.W.); guorui231@mails.ucas.ac.cn (R.G.); ytji@yic.ac.cn (Y.J.); 2Center for Ocean Mega-Science, Chinese Academy of Sciences, 7 Nanhai Road, Qingdao 266071, China; 3University of Chinese Academy of Sciences, Beijing 100049, China; 4School of Basic Sciences for Aviation Naval Aviation University, Yantai 264001, China; xrliang@yic.ac.cn; 5Shandong Saline-Alkali Land Modern Agriculture Company, Dongying 257300, China; dyjwggw@126.com

**Keywords:** antioxidant activity, carboxymethyl chitosan derivatives, quinoline

## Abstract

A total of 16 novel carboxymethyl chitosan derivatives bearing quinoline groups in four classes were prepared by different synthetic methods. Their chemical structures were confirmed by Fourier-transform infrared spectroscopy (FTIR), nuclear magnetic resonance (NMR), and elemental analysis. The antioxidant experiment results in vitro (including DPPH radical scavenging ability, superoxide anion radical scavenging ability, hydroxyl radical scavenging ability, and ferric reducing antioxidant power) demonstrated that adding quinoline groups to chitosan (CS) and carboxymethyl chitosan (CMCS) enhanced the radical scavenging ability of CS and CMCS. Among them, both *N*, *O*-CMCS derivatives and *N*-TM-*O*-CMCS derivatives showed DPPH radical scavenging over 70%. In addition, their scavenging of superoxide anion radicals reached more than 90% at the maximum tested concentration of 1.6 mg/mL. Moreover, the cytotoxicity assay was carried out on L929 cells by the MTT method, and the results indicated that all derivatives showed no cytotoxicity (cell viability > 75%) except *O*-CMCS derivative 1a, which showed low cytotoxicity at 1000 μg/mL (cell viability 50.77 ± 4.67%). In conclusion, the carboxymethyl chitosan derivatives bearing quinoline groups showed remarkable antioxidant ability and weak cytotoxicity, highlighting their potential use in food and medical applications.

## 1. Introduction

Playing essential roles in combating intracellular microorganisms, reactive oxygen species (ROS) are natural byproducts of cellular oxidative metabolism [1,2,3]. However, excessive ROS adversely affects genetic material, proteins, and lipid membranes, leading to cellular senescence, carcinogenesis, and inflammation [4,5]. Therefore, removing excess ROS from the body can effectively delay cellular senescence, inhibit malignant tumors, and prevent inflammation. Research on antioxidants continues to find more efficient solutions for clearing excess reactive oxygen species in the body.

Quinoline, present in natural products, has various biological activities such as antioxidant, antibacterial, anti-inflammatory, and anti-tumor [6,7,8]. Runge isolated quinoline from coal tar in 1834 and named it “Leukol”. Gerhardt officially named it “quinoline” in 1842 [9]. Subsequently, more and more researchers have shown great interest in this nitrogen-containing heterocyclic compound with unique molecular structure and biological activity. Douadi et al. synthesized a series of azoimine quinoline derivatives with excellent antioxidant, anti-inflammatory, and antimicrobial activities [10]. Mahajan’s group prepared a variety of novel thieno[2,3-*b*]quinoline-2-carboxylic acid derivatives, including β-diketone, pyrazole, and flavone. It was found that these derivatives exhibit a strong scavenging ability for the DPPH radical [11]. In addition, various drugs containing quinoline groups have been widely used in treating malaria, inflammation, and cancer. Such as quinine, quinidine, topotecan, and irinotecan.

Chitosan, a deacetylation product of chitin, has a broad application prospect in food packaging due to its antioxidant activity [12,13]. Carboxymethyl chitosan (CMCS) is a vital derivative with better water solubility and biological activity than chitosan [14,15]. Three types of carboxymethyl chitosan can be obtained through different preparation methods: *O*-carboxymethyl chitosan, *N*-carboxymethyl chitosan, and *N*, *O*-carboxymethyl chitosan. CMCS is vital in medical materials, food, environmental protection, and other fields [16,17,18]. Hashmi’s group developed a Tacrolimus-loaded carboxymethyl chitosan medical scaffold with promising antibacterial activity for improving angiogenesis, fibroblast proliferation, and inflammation [19]. Zhao’s team prepared a composite hydrogel containing oxidized pullulan polysaccharide and carboxymethyl chitosan that is injectable, self-healing, antibacterial, and pro-healing, making it suitable for protecting and treating open abdominal wounds [20]. In addition, Liu et al. used a self-assembling composite membrane made of carboxymethyl chitosan and zinc alginate to preserve refrigerated meat. The membrane demonstrated significant water resistance and antibacterial properties [21]. Sela’s research group synthesized derivatives of carboxymethyl chitosan grafted with quercetin through a Schiff base reaction in one step. The derivatives demonstrated excellent antioxidant and antifungal activities. Moreover, they were found to be effective in slowing water loss and the browning of fresh-cut fruits. This discovery holds great importance in the field of food preservation [22]. Zhang et al. developed a sodium alginate/carboxymethyl chitosan composite hydrogel bead with high water absorption that effectively removes methylene blue dye from water bodies [23]. Although researchers’ in-depth study of carboxymethyl chitosan has achieved good research results, carboxymethyl chitosan still has excellent development potential and broad development prospects in antioxidant, antibacterial, and anti-tumor [24,25,26].

In this thesis, CMCS was prepared and slightly modified according to existing reports [27,28,29]. We believe that quinoline groups can serve as the core backbone of multiple drugs with various biological activities. To test this hypothesis, we have systematically introduced quinoline groups into chitosan molecules through acylation reactions for the first time. We have also tested the antioxidant activity of CMCS derivatives. A total of 16 novel carboxymethyl chitosan derivatives bearing quinoline groups in four classes were prepared by different synthetic methods. These included four *O*-carboxymethyl chitosan derivatives bearing quinoline groups, four *N*-carboxymethyl chitosan derivatives bearing quinoline groups, four *N*, *O*-carboxymethyl chitosan derivatives bearing quinoline groups, and four *N*, *N*, *N*-trimethyl-*O*-carboxymethyl chitosan derivatives bearing quinoline groups. Their structures were characterized by FTIR and ^1^H NMR. In addition, their nitrogen and carbon contents were determined by elemental analysis, and their degree of substitution was calculated. In this study, carboxymethyl chitosan derivatives were tested for their antioxidant properties, such as their ability to scavenge DPPH, superoxide anion, and hydroxyl radicals, as well as their ferric-reducing antioxidant power. Additionally, the cytotoxicity of these derivatives to mouse fibroblasts was measured in vitro. The results indicated that most of the derivatives exhibited strong free radical scavenging ability while also showing low cytotoxicity.

## 2. Results and Discussion

### 2.1. Chemical Synthesis and Characterization

Figure 1 outlines the synthetic reaction for creating carboxymethyl chitosan derivatives with quinoline groups. Subsequently, FTIR (Figure 1) and ^1^H NMR (Figure 2) spectroscopy were used to characterize the chemical structures of all carboxymethyl chitosan derivatives. Additionally, Table 1 displays their yields and degrees of substitution (*DS*).

#### 2.1.1. Yields and DS Analysis

The yields and the degrees of substitution of carboxymethyl chitosan derivatives are shown in Table 1. The deacetylation degree of chitosan was 68.67%. The DS values of *O*-CMCS derivatives were 28.35%, 20.70%, 16.38%, and 18.47%, respectively. The DS values of *N*-CMCS derivatives were 9.80%, 14.89%, 18.28%, and 26.25%, respectively. The DS values of *N*, *O*-CMCS derivatives were 15.84%, 15.63%, 15.63%, and 20.57%, respectively. The DS values of *N*-TM-*O*-CMCS derivatives were 16.85%, 3.42%, 9.31%, and 6.81%, respectively. Therefore, the changes in the content of C/N elements proved the success of the modification of CMCS.

#### 2.1.2. FTIR Spectra Analysis

Evidence of chitosan derivative synthesis was explained by FTIR spectroscopy. Figure 1 displays the FTIR spectra of chitosan, carboxymethyl chitosan, and carboxymethyl chitosan derivatives with quinoline groups. In the FTIR spectrum of chitosan, the absorption peak near 3391 cm^−1^ was attributed to the absorption peak generated by the stretching vibration of the O-H bond and N-H bond; the weak broad band peak appearing near 2928 cm^−1^ was attributed to the vibration absorption peak of the C-H bond; and the absorption peak of the C2 amino group of chitosan appeared at 1606 cm^−1^. Moreover, the absorption peak at 1076 cm^−1^ is the stretching vibration peak of the C-O bond [30,31]. In the FTIR spectra of CMCS, the absorption peaks at around 1603, 1409, and 1075 cm^−1^ are the asymmetric stretching vibration peaks of the carboxyl group [32,33]. Furthermore, in the spectrum of *N*-TM-*O*-CMCS, the peak at 1472 cm^−1^ was attributed to the absorption peak of N(CH_3_)_3_^+^ [34,35]. In the FTIR spectra of CMCS derivatives bearing quinoline groups, new absorption peaks appear in the 1647–1653 cm^−1^ attachment and are thought to be characteristic of the amide bond [36]. The characteristic absorption peaks of the quinoline ring appear around 1518–1556 cm^−1^, 1381–1383 cm^−1^, and 1223–1252 cm^−1^ [37,38,39]. In conclusion, the preliminary analysis proves the successful preparation of CMCS derivatives. More structural characteristics require verification by NMR.

#### 2.1.3. ^1^H NMR Spectra Analysis

The correctness of the structure of the carboxymethyl chitosan derivatives bearing quinoline groups was further confirmed by ^1^H NMR spectroscopy. The chemical shifts of the hydrogen atoms [H1], [H2], and [H3]–[H6] in the chitosan molecule are 4.57 ppm, 3.10 ppm, and 3.50–4.01 ppm, respectively [30]. The hydrogen atom on the methylene group of the CMCS molecule has a chemical shift of approximately 3.30 ppm and 3.90 ppm [40]. The peak of the proton signal on the aromatic quinoline ring was observed between 7.10 and 9.50 ppm [41]. Thus, the successful synthesis of carboxymethyl chitosan derivatives bearing quinoline was further demonstrated by the ^1^H NMR data.

### 2.2. Antioxidant Activity

It has been reported that the occurrence of cardiovascular and cerebrovascular diseases such as cancer, premature aging, rheumatoid arthritis, and diabetes may be due to excessive ROS-mediated oxidative stress in the human body, which damages vital substances such as lipids, proteins, and DNA. Therefore, we chose to evaluate the antioxidant activity of CMCS derivatives by testing their ability to scavenge DPPH radical, superoxide anion radical, hydroxyl radical, and ferric reduction ability compared to Vc as a positive control (Figure 3, Figure 4, Figure 5 and Figure 6).

It can be seen from Figure 3 that the chitosan raw material has a weak ability to scavenge DPPH radical, and its scavenging rate is only 26.07 ± 4.47% at the maximum tested concentration of 1.6 mg/mL. The ability of CMCS derivatives to scavenge DPPH radicals is higher than that of both CMCS and CS, and this ability increases with an increase in sample concentration. The *N*-CMCS derivative 2d has the highest scavenging ability due to the retention of the C6 hydroxyl group and a higher degree of substitution. At the tested concentration of 0.4 mg/mL, its DPPH radical scavenging index reached 100%. The clearance rate of all CMCS derivatives is higher than 50% at the maximum tested concentration of 1.6 mg/mL. In particular, the *N*-TM-*O*-CMCS derivatives exhibit the most significant improvement compared to chitosan. Additionally, all other derivatives exhibit a scavenging effect exceeding 85%, except for the derivative 4a.

**Figure 3 marinedrugs-21-00606-f003:**
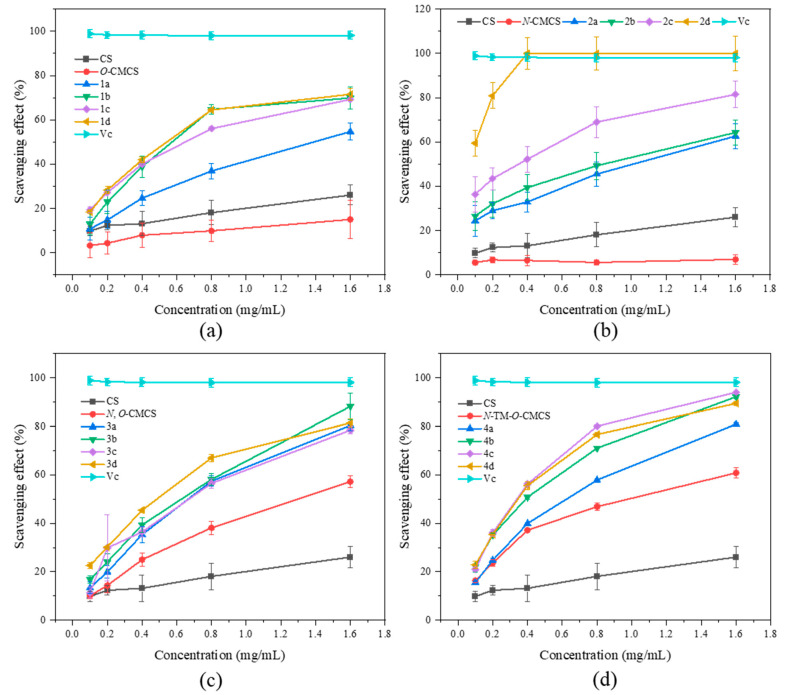
DPPH radical scavenging activity of chitosan and CMCS derivatives ((**a**): *O*-CMCS derivatives, (**b**): *N*-CMCS derivatives, (**c**): *N*, *O*-CMCS derivatives, (**d**): *N*-TM-*O*-CMCS derivatives).

In Figure 4, the superoxide anion-scavenging ability of CMCS derivatives is demonstrated. The scavenging capacity of all samples was found to increase with concentration. The scavenging ability of chitosan on superoxide anion-free radicals is significantly better than the scavenging ability of DPPH-free radicals. At a concentration of 1.6 mg/mL, chitosan shows a scavenging ability of 57.82 ± 1.79%. At the highest concentration tested (1.6 mg/mL), all derivatives, except *O*-CMCS derivatives, exhibited over 80% scavenging ability against superoxide anion radicals. In particular, the *N*-CMCS and *N*-TM-*O*-CMCS derivatives exhibit a scavenging ability that surpasses 90%. The results indicate that adding quinoline groups enhances CS’s ability to scavenge superoxide anion radicals.

**Figure 4 marinedrugs-21-00606-f004:**
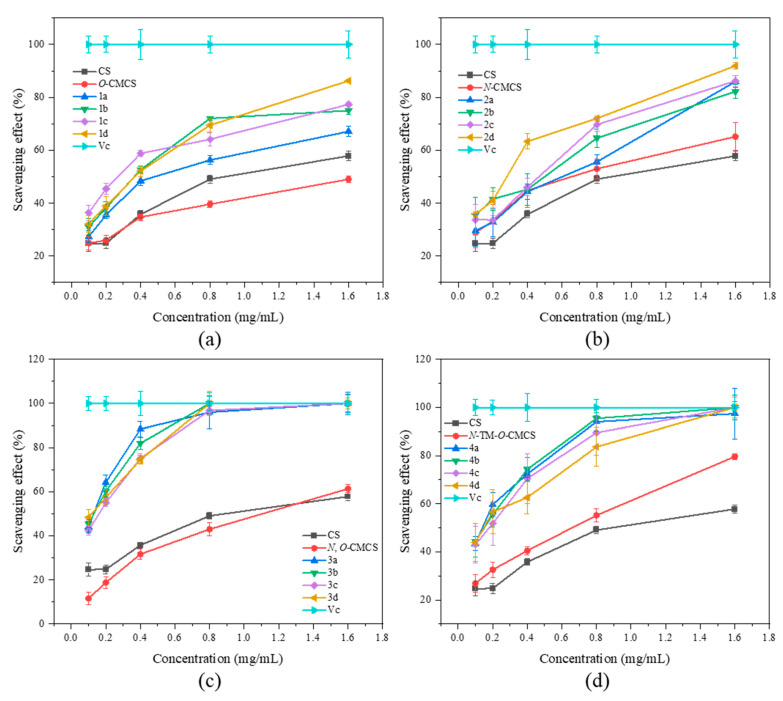
Superoxide anion radical scavenging activity of chitosan and CMCS derivatives ((**a**): *O*-CMCS derivatives, (**b**): *N*-CMCS derivatives, (**c**): *N*, *O*-CMCS derivatives, (**d**): *N*-TM-*O*-CMCS derivatives).

In Figure 5, it is demonstrated that CS, CMCS, and CMCS derivatives have the ability to scavenge hydroxyl radicals. Additionally, the scavenging ability of all the samples is directly proportional to their concentration. Furthermore, the scavenging ability of CMCS derivatives is found to be superior to that of both CS and CMCS. The derivatives have a slightly weaker ability to scavenge hydroxyl radicals than their ability to scavenge DPPH radicals and superoxide anion radicals. However, *N*, *O*-CMCS derivative 2b has a clearance rate of 85.24 ± 4.39% even at the maximum tested concentration of 1.6 mg/mL, while the clearance rates of other derivatives are less than 80%. Among the derivatives tested, *N*-TM-*O*-CMCS derivatives (4a–4d) demonstrated the highest ability to scavenge hydroxyl radicals, with values exceeding 50%. Specifically, the values were 74.71 ± 3.38%, 69.71 ± 3.26%, 59.06 ± 3.21%, and 57.49 ± 3.49%, respectively.

**Figure 5 marinedrugs-21-00606-f005:**
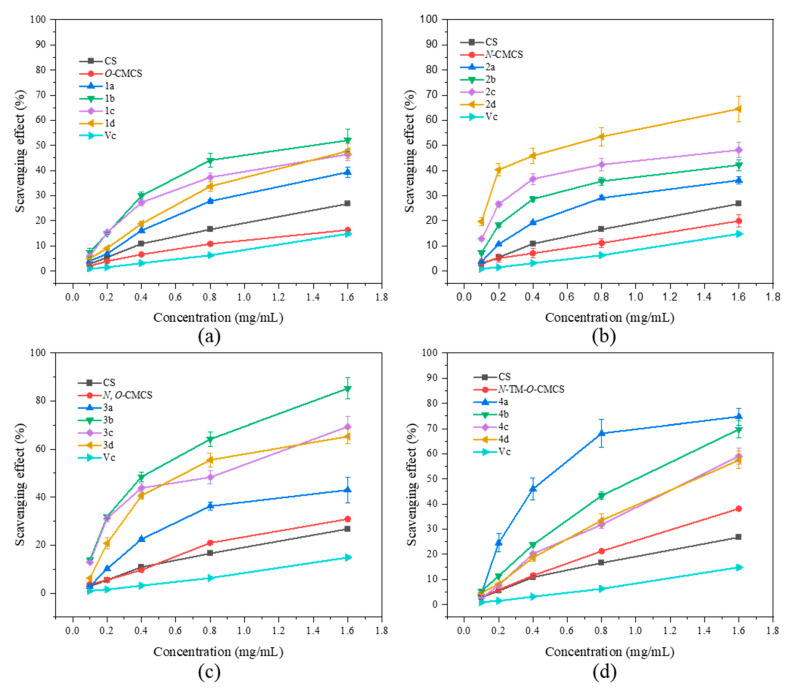
Hydroxyl radical scavenging activity of chitosan and CMCS derivatives ((**a**): *O*-CMCS derivatives, (**b**): *N*-CMCS derivatives, (**c**): *N*, *O*-CMCS derivatives, (**d**): *N*-TM-*O*-CMCS derivatives).

As shown in Figure 6, none of the CMCS derivatives show a significant increase in ferric-reducing antioxidant power. Among them, the *O*-CMCS derivative 2b demonstrates the highest reducing power, but it is only 1.53 ± 0.03 A, while all the other CMCS derivatives are all below 1.5 A. This suggests that adding quinoline groups has little effect on improving the ferric-reducing antioxidant power of CS.

**Figure 6 marinedrugs-21-00606-f006:**
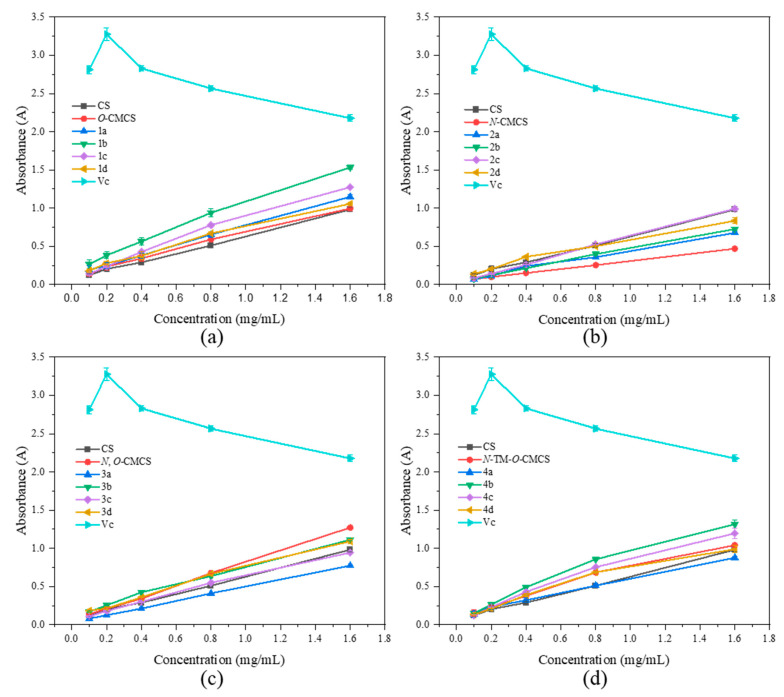
Ferric-reducing antioxidant power of chitosan and CMCS derivatives ((**a**): *O*-CMCS derivatives, (**b**): *N*-CMCS derivatives, (**c**): *N*, *O*-CMCS derivatives, (**d**): *N*-TM-*O*-CMCS derivatives).

### 2.3. Cytotoxicity Analysis

Figure 7 shows the bar chart of the cell survival rate of L929 cells cultured with CS, CMCS, and CMCS derivatives for 24 h. Moreover, Figure 8 shows the cell morphology of L929 cells after 24 h of incubation in a sample solution at 1000 μg/mL. Except for *O*-CMCS and its derivatives 1a at high concentrations (1000 μg/mL), which show high cytotoxicity (cell survival rates of 58.46 ± 11.64% and 50.77 ± 4.67%), other derivatives exhibit low toxicity or no cytotoxicity. *O*-CMCS derivatives (1b–1d): 109.67 ± 5.67%, 99.73 ± 4.95%, and 105.46 ± 2.98%; *N*-CMCS derivatives (2a–2d): 94.51 ± 4.85%, 88.91 ± 9.78%, 78.58 ± 7.21%, and 100.32 ± 10.27%; *N*, *O*-CMCS derivatives (3a–3d): 91.16 ± 2.10%, 98.92 ± 1.57%, 82.48 ± 2.78%, and 96.01 ± 5.60%; *N*-TM-*O*-CMCS derivatives (4a–4d): 114.08 ± 2.53%, 84.90 ± 6.01%, 81.32 ± 7.28%, and 101.15 ± 5.69%. In conclusion, the introduction of quinoline groups has little effect on the biocompatibility of CS. Therefore, CMCS derivatives bearing quinoline groups have a certain application potential as synthetic antioxidants.

## 3. Materials and Methods

### 3.1. Materials

Golden-Shell Pharmaceutical Co., Ltd. (Zhejiang, China) supplied chitosan with a molecular weight of 5000–8000 Da and 68.67% deacetylation degree. Furthermore, 3-aminoquinoline, 5-aminoquinoline, 6-aminoquinoline, and 8-aminoquinoline were purchased from Sigma-Aldrich Chemical Corp. (Shanghai, China). 1-(3-Dimethylaminopropyl)-3-ethylcarbodiimide hydrochloride (EDC) and *N*-hydroxysuccinimide (NHS) were purchased from Shanghai Macklin Biochemical Co., Ltd. (Shanghai, China). Sodium hydroxide, glyoxylic acid monohydrate, isopropanol, chloroacetic acid, hydrochloric acid, sodium iodide, iodomethane, dimethyl sulfoxide (DMSO), acetone, ethanol, and *N*-methylpyrrolidone (NMP) were provided by Sinopharm Chemical Reagent Co., Ltd. (Shanghai, China). The reagents used in the experiment in this paper are analytically pure, and the experimental water is deionized water.

### 3.2. Preparation of Chitosan Derivatives

#### 3.2.1. Synthesis of *O*-Carboxymethyl Chitosan (*O*-CMCS)

One gram of chitosan (6.2 mmol) was accurately weighed, and 10 mL of isopropyl alcohol was placed in a 100 mL reaction flask. Then, 2.5 mL of a 40% NaOH solution was slowly added dropwise, and the reaction was performed at room temperature for 1 h. Then, 23 mL of a 10% chloroacetic acid solution was added drop by drop. The reaction was performed at room temperature for 4 h. After the reaction, the system’s pH value was adjusted to 5–6 with hydrochloric acid, and the system was poured into excess ethanol to precipitate a large amount of light yellow solid. The system underwent filtration and dialysis in deionized water for 48 h using a 500 Da cut-off molecular-weight dialysis bag. Subsequently, *O*-CMCS was obtained via vacuum freeze-drying.

#### 3.2.2. Synthesis of *N*-Carboxymethyl Chitosan (*N*-CMCS)

To synthesize *N*-CMCS, 2.0 g of chitosan (12.4 mmol) was dissolved in 20 mL of deionized water. Then, a solution containing 8.17 g of glyoxylic acid monohydrate (88.8 mmol) was added slowly. The reaction was performed at room temperature for 2.5 h. Next, NaOH was added to make the system’s pH 10, and 4.7 g of sodium borohydride (124 mmol) was added in batches. The reaction continued at room temperature for 5 h. Once the reaction was complete, hydrochloric acid was added to make the system pH 5. The next step involved dialysis in deionized water for 48 h using a molecular weight interception in a 500 Da dialysis bag. Finally, *N*-CMCS (white solid) was obtained after vacuum freeze-drying.

#### 3.2.3. Synthesis of *N*, *O*-Carboxymethyl Chitosan (*N*, *O*-CMCS)

Four grams of chitosan (24.8 mmol) were added to 40 mL of isopropanol, and 10 mL of a 40% NaOH solution was added for 1 h. The reaction was heated to 60 °C, and 7.5 g of chloroacetic acid (79.4 mmol) was added, followed by a 5-hour reaction. At the end of the reaction, deionized water was added to dilute the system. The solution was then poured into excess ethanol and filtered to obtain a yellow solid. This solid was washed with ethanol three times and vacuum freeze-dried to obtain *N*, *O*-CMCS.

#### 3.2.4. Synthesis of *N*, *N*, *N*-Trimethyl-*O*-Carboxymethyl Chitosan (*N*-TM-*O*-CMCS)

First, 1.61 g of chitosan (10 mmol) was dissolved in 40 mL of NMP, and 4.5 g of NaI (30 mmol), 15 mL of a 15% NaOH solution, and 15 mL of CH_3_I were slowly added under an ice bath. Then, the reaction was heated to 60 °C and refluxed for 2 h. After cooling to room temperature, the reaction solution was poured into excess ethanol and filtered to obtain a large amount of solid. The solid was dried in a vacuum and then dissolved in 10 mL of isopropanol. Next, 2.5 mL of a 40% NaOH solution was added drop by drop, and the reaction was allowed to proceed for 1 h. After that, 23 mL of a 10% chloroacetic acid solution was slowly added to the reaction at room temperature for 6 h. Excess acetone was added once the reaction was complete, and the system was filtered to obtain a brown solid. The filter cake was washed three times with acetone and dried under vacuum to obtain *N*-TM-*O*-CMCS.

#### 3.2.5. Synthesis of Carboxymethyl Chitosan Derivatives Bearing Quinoline Groups (1a–1d, 2a–2d, 3a–3d, and 4a–4d)

According to the preparation method of *O*-carboxymethyl chitosan derivatives bearing quinoline groups (1a–1d), the rest of the carboxymethyl chitosan derivatives bearing quinoline groups (2a–2d, 3a–3d, and 4a–4d) were prepared in the same way. First, 1.0 g of *O*-CMCS (4.5 mmol), 1.0 g of EDC (4.5 mmol), and 0.55 g of NHS (4.5 mmol) were added to the reaction flask containing 20 mL of DMSO. Concentrated hydrochloric acid was added to make the system clear and transparent, and nitrogen was used to displace the air. The mixture was then allowed to react for 5 h at room temperature without light. Next, a solution of 1.97 g of aminoquinoline (13.7 mmol) in 20 mL of DMSO was added drop by drop. The reaction was conducted without light for 10 h at room temperature in a nitrogen atmosphere. After the reaction was completed, the reaction system was poured into excess acetone, filtered to obtain a large number of solids, and the filter cake was extracted by Soxhlet using ethanol for 48 h to obtain *O*-carboxymethyl chitosan derivatives bearing quinoline groups (1a–1d).

### 3.3. Analytical Methods

#### 3.3.1. Fourier Transform Infrared Spectroscopy (FTIR)

FTIR analysis was carried out on a Nicolet iS 50 Fourier Transform Infrared Spectrometer (Thermo, Waltham, MA, USA), using transmittance modes at a resolution of 4.0 cm^−1^ in the 4000–500 cm^−1^ region. The tested samples were treated with the potassium bromide table method for observation with the accumulation of 16 scans at room temperature.

#### 3.3.2. ^1^H Nuclear Magnetic Resonance Spectroscopy (^1^H NMR)

^1^H NMR was carried out on a Bruker AVIII-500 Spectrometer (Switzerland, provided by Bruker Tech. and Serv. Co., Ltd., Beijing, China) operating at 500 MHz to determine the chemical structures of the prepared carboxymethyl chitosan derivatives dissolved in D_2_O or DMSO. The detection temperature was 25 °C. 

#### 3.3.3. Degrees of Substitution (DS)

The elemental composition of the prepared samples was performed on a Vario EL III elemental analyzer. According to the ratio of carbon and nitrogen content, the degree of deacetylation (*DD*) of chitosan, the degree of substitution of intermediate products (*DS*_1_), and the degree of substitution of carboxymethyl chitosan derivatives (*DS*_2_) were calculated as follows:$$DD=\frac{n_{1}\times M_{C}-M_{N}\times W_{C/N}}{n_{2}\times M_{C}}$$
$${DS}_{1}=\frac{W_{C/N}\times M_{N}-n_{1}\times M_{C}+n_{2}\times DD\times M_{C}}{n_{3}\times M_{C}}$$
$${DS}_{2}=\frac{W_{C/N}\times M_{N}-n_{1}\times M_{C}+n_{2}\times DD\times M_{C}-n_{3}\times {DS}_{1}\times M_{C}}{n_{4}\times M_{C}-n_{1}^{*}\times W_{C/N}\times M_{N}}$$
where *n*_1_, *n*_2_, *n*_3_, *n*_4_, and $n_{1}^{*}$ represent the number of carbon atoms in the chitin molecule, the number of carbon atoms in the acetyl group, the number of carbon atoms remaining in the chitosan molecule in the intermediate product, the number of carbon atoms in aminoquinoline, and the number of nitrogen atoms in aminoquinoline, *n*_1_ = 8, *n*_2_ = 2, *n*_3_ = 2 or 4 or 5, respectively. *n*_4_ = 9 or 18, $n_{1}^{*}$ = 2 or 4; *M_C_* and *M_N_* represent the relative atomic masses of carbon and nitrogen, *M_C_* = 12, *M_N_* = 14. *W_C/N_* represents the ratio of carbon to nitrogen content in the sample. 

### 3.4. Antioxidant Assay In Vitro

#### 3.4.1. DPPH Radical Scavenging Activity

The DPPH radical scavenging ability of chitosan, carboxymethyl chitosan, and carboxymethyl chitosan derivatives was measured following Zhang’s method with slight modifications [42]. All tested samples (chitosan, carboxymethyl chitosan, and carboxymethyl chitosan derivatives) were prepared in a series of 1 mL aqueous solutions of 0.3, 0.6, 1.2, 2.4, and 4.8 mg/mL, respectively. The reaction mixture, involving 1 mL of the test samples and 2 mL of DPPH-ethanol solution (180 μM), was incubated in the dark at room temperature for 20 min. To prepare the blank solution, 1.0 mL of deionized water was added instead of 1.0 mL of sample solution. Absolute ethanol was used as the control instead of the DPPH-ethanol solution. The DPPH radical scavenging activity was evaluated by measuring the absorbance at 517 nm. A triplicate measurement was taken for each test sample, and the scavenging rate of DPPH radical was calculated according to the following formula: $$Scavenging\;effect\;(\%)=\left[1-\frac{A_{sample\;517nm}-A_{control\;517nm}}{A_{blank\;517nm}}\right]\times 100$$
where *A_sample_*
_517nm_ represents the absorbance of samples at 517 nm, *A_control_*
_517nm_ represents the absorbance of the control at 517 nm, and *A_blank_*
_517nm_ represents the absorbance of the blank at 517 nm.

#### 3.4.2. Superoxide Anion Radical Scavenging Activity

The superoxide anion radical scavenging activity assay of chitosan, carboxymethyl chitosan, and carboxymethyl chitosan derivatives was conducted according to the previous method with a minor adjustment [43]. Firstly, 36.57 mg reduced coenzyme I (NADH), 24.53 mg nitroblue tetrazolium (NBT), and 1.838 mg phenazine methyl sulfate (PMS) were weighed and dissolved in 100 mL of Tris-HCl buffer (16 mM, pH 8.2), respectively. All tested samples (chitosan, carboxymethyl chitosan, and carboxymethyl chitosan derivatives) were dissolved in deionized water and prepared into a series of 1.5 mL aqueous solutions of 0.20, 0.40, 0.80, 1.60, and 3.20 mg/mL. Subsequently, 0.50 mL of NADH, NBT, and PMS were added to various sample solutions. The mixture was evenly mixed and reacted for 5 min at room temperature in the dark. The absorbance was measured at 560 nm after the reaction. The blank group was replaced with deionized water, and the control group’s buffer was substituted with NADH. All experiments were conducted in triplicate, and the effects of scavenging superoxide anion radicals were calculated using the following formula:$$Scavenging\;effect\;(\%)=\left[1-\frac{A_{sample\;560nm}-A_{control\;560nm}}{A_{blank\;560nm}}\right]\times 100$$
where *A_sample_*
_560nm_ is the absorbance of the sample at 560 nm, *A_control_*
_560nm_ is the absorbance of the blank group at 560 nm, and *A_blank_*
_560nm_ is the absorbance of the control group at 560 nm.

#### 3.4.3. Hydroxyl Radical Scavenging Activity

Hydroxyl radical scavenging activity was performed using the previous method with slight modifications [44]. Firstly, 1.0 mL of samples at different concentrations (0.45, 0.90, 1.80, 3.60, and 7.20 mg/mL) were mixed with 0.5 mL of EDTA-Fe^2+^ (220 μM). Then, 2.0 mL of phosphate buffer solution (pH = 7.4) containing safranine T (0.23 μM) and H_2_O_2_ (60 μM) was added to a final volume of 3.0 mL. Instead of using the sample solutions, 1.0 mL of deionized water was used for the blank group, and 1.0 mL of phosphate buffer solution was used for the control group instead of 3% hydrogen peroxide solution. The reaction mixture was shaken quickly and incubated in the dark for 30 min at 37 °C. Three replicates for every sample concentration were performed, and the absorbance was measured at 520 nm. To calculate the rate of hydroxyl radical scavenging, the following formula was used:$$Scavenging\;effect\;(\%)=\left[\frac{A_{sample\;520nm}-A_{blank\;520nm}}{A_{control\;520nm}-A_{blank\;520nm}}\right]\times 100$$
where *A_sample_*
_520nm_ is the absorbance of the sample group, *A_control_*
_520nm_ is the absorbance of the control group, and *A_blank_*
_520nm_ is the absorbance of the blank group.

#### 3.4.4. Ferric-Reducing Antioxidant Power

The ferric-reducing antioxidant power was conducted following the procedure outlined by Li [45]. Samples at different concentrations (0.60, 1.20, 2.40, 4.80, and 9.60 mg/mL) were mixed with potassium ferricyanide (1%, *w*/*v*) and incubated at 50 °C for 20 min. One milliliter of trichloroacetic acid solution (10%, *w*/*v*) was added after the reaction mixture was cooled to room temperature. After shaking, the mixture was centrifuged (3000 r/min) for 5 min to obtain the supernatant. Furthermore, 1.5 mL of the supernatant was mixed with 1.2 mL of deionized water and 0.3 mL of ferric chloride (0.1%, *w*/*v*). The mixture was shaken quickly and reacted for 10 min at 25 °C. All samples and the blank (deionized water) were performed in triplicate and measured at 700 nm. A higher absorbance value indicates a more substantial reduction in power.

### 3.5. Cytotoxicity Assay

#### 3.5.1. Cell Preparation and Culturing

Briefly, L929 cells (fibroblasts) were cultured in Dulbecco’s modified Eagle’s medium (DMEM) supplemented with 10% fetal bovine serum at 37 °C and 5% CO_2_.

#### 3.5.2. Cell Viability Assay

The cytotoxicity of chitosan, carboxymethyl chitosan, and carboxymethyl chitosan derivatives was investigated using the MTT assay [46]. L929 cells were cultured to the exponential growth phase and diluted into cell suspension at a density of 5–10 × 10^4^ cells/mL. Then, 100 μL of cell suspension was added to sterile 96-well plates and incubated at 37 °C with 5% CO_2_ for 24 h. The samples with different final concentrations were added to the 96-well plates, and the cells were cultured for another 24 h. A total of five parallel tests were established for each sample concentration. Next, the culture medium was removed, and 100 μL of MTT solution was added to each well and incubated for 4 h at 37 °C. Finally, 150 μL of dimethyl sulfoxide (DMSO) was added to dissolve the crystals, and the absorbance of each well was measured at 490 nm. Cell viability was calculated using the following formula:$$Cell\;viability\;\left(\%\right)=\frac{A_{sample\;490\;nm}-A_{blank\;490\;nm}}{A_{control\;490\;nm}-A_{blank\;490\;nm}}\times 100$$
where *A_sample_*
_490 nm_ is the absorbance of the samples at 490 nm, *A_blank_*
_490 nm_ is the absorbance of the blank at 490 nm, and *A_control_*
_490 nm_ is the absorbance of the negative control at 490 nm.

### 3.6. Statistical Analysis

All experiments related to antioxidant activity and cytotoxicity were performed in triplicate. Data were reported as the mean ± standard deviation (SD) and analyzed by one-way analysis of variance. Significant differences (*p* < 0.05) between the means were determined using Scheffe’s multiple range test. The data figures were prepared using OriginPro 2021 software (OriginLab Corporation, Northampton, MA, USA). 

## 4. Discussion

According to the in-depth study, antioxidants provide significant protection against oxidative stress in various pathological processes. In addition, with the development of the economy, people have higher requirements for food preservation technology, and synthetic antioxidants are usually highly toxic and unsuitable as food additives and packaging materials. Chitosan and its derivatives, as natural polymers, have attracted much attention in the field of antioxidants because of their non-toxic and easy degradation characteristics. In this paper, the antioxidant activity of CMCS derivatives containing quinoline groups was preliminarily determined using different methods. The results showed that the prepared CMCS derivatives had good free radical scavenging ability, especially the *N*, *O*-carboxymethyl chitosan derivatives and *N*-TM-*O*-CMCS derivatives, which were generally better than *O*-CMCS derivatives and *N*-CMCS derivatives, which might be attributed to the unique structure of the quinoline ring. Its ring nitrogen atom has a pair of lone electrons, which can be used as an electron donor to convert reactive free radicals into stable products that quench free radicals. Therefore, modification of the amino group and hydroxyl group of chitosan may be more beneficial to improve its free radical scavenging ability. In addition, cytotoxicity experiments also preliminarily demonstrated that the prepared CMCS derivatives were safe and non-toxic. Although the preliminary experimental results can prove that CMCS derivatives have good antioxidant capacity and biocompatibility, which can provide theoretical support for applying CMCS derivatives in the food and pharmaceutical fields, more in-depth experiments are still needed to evaluate their antioxidant activity and cytotoxicity.

## 5. Conclusions

In this paper, 16 novel CMCS derivatives were successfully synthesized by EDC/NHS catalysis. Their chemical structures were characterized by FTIR, ^1^H NMR, and elemental analysis to confirm the successful introduction of the quinoline groups. Currently, there are only four in vitro antioxidant models available for testing the antioxidant activities of derivatives of carboxymethyl chitosan (CMCS). However, these models are still in the early experimental stage. To confirm the potential and application scope of CMCS derivatives as synthetic antioxidants in the future, more comprehensive research techniques are required. Additionally, the cytotoxic results suggested that the introduction of quinoline groups had a minimal impact on the biocompatibility of CS and CMCS. In summary, the CMCS derivatives synthesized in this study demonstrated outstanding antioxidant activity and biocompatibility. Ongoing studies are being conducted in our laboratory to explore the synthetic applications of these derivatives.

## Data Availability

All data contained in the manuscript are available from the authors.

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
