# Peer review of "Preparation and Antioxidant Activity of New Carboxymethyl Chitosan Derivatives Bearing Quinoline Groups"

_marinedrugs, 2023, doi:10.3390/md21120606_

Round 1

Reviewer 1 Report

Comments and Suggestions for Authors

The preparation of new highly active, non-toxic, biocompatible antioxidants is an important direction in modern chemistry, materials science, pharmacology and food bioscience. The authors correctly identified the problem. Chitosan can help a lot in solving this problem. The authors offer an original approach. This paper focuses on the preparation of carboxymethyl derivatives of chitosan with different types of substitution followed by the introduction of quinoline substituents through conjugation with the carboxyl group of the carboxymethyl derivative of chitosan via carbodiimide chemistry. The authors carefully characterized the resulting derivatives using a variety of analytical methods, and also assessed the antioxidant activity and toxicity of the resulting polymers. The article is written very well, competently and logically, well illustrated, provided with the necessary references, the conclusions are logical and do not contradict the data previously described in the literature. The experiment was carried out methodically correctly. This article will be of interest to many readers and will be well cited. However, I recommend minor revision to improve the article.

1. I kindly ask the authors to improve the quality of the drawings, because many of the numbers are hard to see

2. I also ask the authors to provide a method for determining the molecular weight of chitosan and its degree of deacetylation.

3. To enhance the introduction of the part, I suggest quoting 10.1016/j.foodchem.2020.128676 (antioxidant active chitosan based films) and 10.1016/j.foodhyd.2023.109104

Reviewer 2 Report

Comments and Suggestions for Authors

In this study, 16 new carboxymethyl chitosan derivatives bearing quinoline groups in 4 classes were prepared by different synthetic methods. Their chemical structures were confirmed by Fourier-transform infrared spectroscopy (FTIR), nuclear magnetic resonance (NMR), and elemental analysis. The in vitro antioxidant experiment results demonstrated that adding quinoline groups to chitosan (CS) and carboxymethyl chitosan (CMCS) enhanced the radical scavenging ability of CS and CMCS. Although it is a comprehensive study that lots of experiments have been performed, the novelty of the manuscript is limited, which can not meet the requirement of the high standard of the journal. However, if the authors can perform more in-depth experiments, this manuscript may be resubmitted and considered for publication. Here are some suggestions for further revisions.

1, More key results can be provided in the abstract.

2, The introduction section can be improved. For example, the synthesis methods for the preparation of CMCS derivatives can be introduced, are there any new methods used in the present study? Are there any innovation points except for 16 CMCS were obtained?

3, Are there any more analytical methods can be used for the characterization of the chemical structures of CMCS?

4, There were several in vitro models applied in the evaluation of antioxidant activity of CMCS, it is quite preliminary experiment, what is the application potential of CMCS? Are there any other models being considered for the evaluation of the antioxidant activity of CMCS?

5, The quality of figures should be improved greatly. For example, The Figure 1 and 2 are very poor in resolution.

6, The “2.1.3. Yields and DS analysis” can be moved to before “2.1.1. FTIR spectra analysis”.

7, It is suggested to add a discussion section to discuss in-depth and comprehensively on the results of present work.

Round 2

Reviewer 2 Report

Comments and Suggestions for Authors

This manuscript has been carefully revised according to the suggestions. Here are some minor comments for further revisions.

1, There should be a space between the numeral and units.

2, The novelty of present work can be emphasized in the last paragraph of introduction section.

3, Table 1, the line under “CS” can be removed.

4, The resolution of figures can be further improved.

5, The “concentration” in the figures can be changed to “Concentration”.

6, The shortcomings of present work can be emphasized in the conclusion section.

Author Response

Response to Reviewer 2 Comments

Point 1: There should be a space between the numeral and units.

Response 1: We really appreciate your suggestion and have put a space between the number and unit (lines 28, 250, 399).

Point 2: The novelty of present work can be emphasized in the last paragraph of introduction section.

Response 2: Thank you for your kind suggestions. We have emphasized the novelty of present work in the last paragraph of introduction section (lines 79-82).

Point 3: Table 1, the line under “CS” can be removed.

Response 3: Thank you for your kind suggestions. We have removed the line under "CS" in Table 1 (line 105).

Point 4: The resolution of figures can be further improved.

Response 4: Thank you for your kind suggestions. We have modified the resolution of figures (lines 150, 174, 186, 201, 209, 225, 228 and 233).

Point 5: The “concentration” in the figures can be changed to “Concentration”.

Response 5: We really appreciate your suggestion and have modified “concentration” in the figures to “Concentration”.

Point 6: The shortcomings of present work can be emphasized in the conclusion section.

Response 6: Thank you for your kind suggestions. We have emphasized the shortcomings of present work in the conclusion section (lines 443-452).